# Impact of the Oral Administration of Polystyrene Microplastics on Hepatic Lipid, Glucose, and Amino Acid Metabolism in C57BL/6Korl and C57BL/6-Lep^em1hwl^/Korl Mice

**DOI:** 10.3390/ijms25094964

**Published:** 2024-05-02

**Authors:** Yujeong Roh, Jieun Kim, Heejin Song, Ayun Seol, Taeryeol Kim, Eunseo Park, Kiho Park, Sujeong Lim, Suha Wang, Youngsuk Jung, Hyesung Kim, Yong Lim, Daeyoun Hwang

**Affiliations:** 1Department of Biomaterials Science (BK21 FOUR Program), Life and Industry Convergence Research Institute, Laboratory Animal Resources Center, College of Natural Resources & Life Science, Pusan National University, Miryang 50463, Republic of Korea; buzyu99@naver.com (Y.R.); prettyjiunx@naver.com (J.K.); hejin1544@naver.com (H.S.); a990609@naver.com (A.S.); xofuf0701@naver.com (T.K.); geg9393@naver.com (E.P.); puji0918@naver.com (K.P.); soojl1315@naver.com (S.L.); dhkdtngk@naver.com (S.W.); 2College of Pharmacy, Pusan National University, Busan 46241, Republic of Korea; 3Department of Nanomechatronics Engineering, College of Nanoscience & Nanotechnology, Pusan National University, Miryang 50463, Republic of Korea; hsk@pusan.ac.kr; 4Department of Clinical Laboratory Science, College of Nursing and Healthcare Science, Dong-Eui University, Busan 47340, Republic of Korea; yonglim@deu.ac.kr

**Keywords:** microplastics, lipogenesis, lipolysis, glycogenolysis, insulin resistance, metabolites

## Abstract

The impact of microplastics (MPs) on the metabolic functions of the liver is currently unclear and not completely understood. To investigate the effects of the administration of MPs on the hepatic metabolism of normal and obese mice, alterations in the lipid, glucose (Glu), and amino acid regulation pathways were analyzed in the liver and adipose tissues of C57BL/6Korl (wild type, WT) or C57BL/6-Lep^em1hwl^/Korl mice (leptin knockout, Lep KO) orally administered polystyrene (PS) MPs for 9 weeks. Significant alterations in the lipid accumulation, adipogenesis, lipogenesis, and lipolysis pathways were detected in the liver tissue of MP-treated WT and Lep KO mice compared to the vehicle-treated group. These alterations in their liver tissues were accompanied by an upregulation of the serum lipid profile, as well as alterations in the adipogenesis, lipogenesis, and lipolysis pathways in the adipose tissues of MP-treated WT and Lep KO mice. Specifically, the level of leptin was increased in the adipose tissues of MP-treated WT mice without any change in their food intake. Also, MP-induced disruptions in the glycogenolysis, Glu transporter type 4 (GLUT4)-5′ AMP-activated protein kinase (AMPK) signaling pathway, levels of lipid intermediates, and the insulin resistance of the liver tissues of WT and Lep KO mice were observed. Furthermore, the levels of seven endogenous metabolites were remarkably changed in the serum of WT and Lep KO mice after MP administrations. Finally, the impact of the MP administration observed in both types of mice was further verified in differentiated 3T3-L1 adipocytes and HepG2 cells. Thus, these results suggest that the oral administration of MPs for 9 weeks may be associated with the disruption of lipid, Glu, and amino acid metabolism in the liver tissue of obese WT and Lep KO mice.

## 1. Introduction

The adverse effects of microplastics (MPs) on hepatic metabolism have been a topic of significant research interest because the liver plays a central role in the metabolism and detoxification of foreign substances [1]. The liver is an essential organ that governs whole-body energy metabolism through metabolic pathways involving glucose (Glu), lipids, and amino acids [2,3]. The liver supplies Glu by converting glycogen into Glu in a process called glycogenolysis. This organ also can produce the necessary Glu by using amino acids and fat byproducts in a process called gluconeogenesis [4,5,6]. Also, the liver contributes to the regulation of lipid metabolism, including the uptake, synthesis, esterification, oxidation, and export of fatty acids [7]. Furthermore, the liver plays a central role in amino acid metabolism, including protein oxidation and its breakdown into amino acids and dipeptides, as well as its conversion to Glu for energy [8,9]. In particular, the liver is a suitable site for assessing the synergistic damage between MPs with high absorbability and various chemicals, including heavy metal ions, bisphenol A, and phthalates, because most of these external materials are primarily transferred to and accumulated in the liver tissue [10,11,12].

Meanwhile, most of the findings show a link between an exposure to MPs of various sizes and a disruption of the hepatic metabolism, although there are differences in the response patterns and the types of MPs. Significant alterations in lipid profile, inflammation, lipid droplets, and endogenous metabolites were detected in ICR mice administered MPs (5 μm and 20 μm) for 28 days, while, in another study, alterations in body and fat weight, lipid profile, and lipogenesis were observed in the same mice treated with MPs (0.5 μm and 50 μm) for 5 weeks [13,14]. Also, the administration of MPs to pregnant female mice for 6 weeks induced changes in the hepatic transcriptome, serum metabolites, and hepatic lipid accumulation in F1 offspring mice [15]. A positive correlation between an MPs treatment and triglyceride (TG) levels was observed in mice administered MPs (5 μm and 20 μm) orally for 28 days [16]. Furthermore, alterations in plasma Glu concentrations and the insulin receptor downstream signaling pathway, without an effect on insulin secretion, were detected after an MPs treatment for 21 days, while the impairment of Glu tolerance, hepatic lipid deposition, and hepatic transcriptional profiles were analyzed in MP-treated C57BL/6 mice after 8 weeks [17,18]. Moreover, the impact of MPs was investigated in C57BL/6 mice with high-fat diet (HFD)-induced obesity. A reduction in lipolysis, defect in their fasting-stimulated lipid mobilization, increase in adipocyte size, and enhancement of lipid accumulation were detected in mice exposed to MPs and an HFD for 8 weeks [19]. Similar alterations including increases in blood Glu concentration, serum lipid levels, nonalcoholic fatty liver disease (NAFLD) activity score, and long-chain fatty acid transportation were detected in HFD-fed mice after a carboxyl-modified fluorescent MPs treatment [20]. However, conflicting findings have reported that MPs of small sizes (less than 1 μm) do not affect many liver metabolism pathways. The concentrations of TG, total cholesterol (TC), and insulin secretion were constantly maintained in mice after treatments with MPs (80 nm) for 21 days (5 mg/kg and 15 mg/kg) and MPs (1 μm) for 8 weeks (100 μg/L and 1000 μg/L) [17,18]. Therefore, additional scientific evidence is required to clarify the issue of the conflicting effects of MPs of small sizes on liver metabolism.

The current study analyzed the molecular mechanisms of MPs of a small size (500 nm) on hepatic lipid, Glu, and amino acid metabolism in C57BL/6Korl (wild type, WT) and C57BL/6-Lep^em1hwl^/Korl (leptin knockout, Lep KO) mice to obtain additional evidence regarding the impact of MPs on liver function. The results of the current study indicate that exposure to MPs of a small size is probably a novel cause for the disruption of lipid, Glu, and amino acid metabolism in the liver tissue of normal and obese mice.

## 2. Results

### 2.1. Accumulation of MPs in the Tissues

First, we investigated whether MPs, which were orally administered for 9 weeks, were present in the liver tissue. For this, the fluorescence intensity of the MPs was observed in a frozen section of liver under a fluorescence microscope. MP particles with an average size of 0.5 μm were detected in the liver section, as shown in Figure 1. These results indicate that, after their administration for 9 weeks, MP particles were found to be deposited in the liver tissue.

### 2.2. Effect of MP Administrations on the Body and Liver Weights of WT and Lep KO Mice

To investigate whether the oral administration of MPs had any effects on the body weight and weight of the liver of normal and obese mice, these weights were measured in WT and Lep KO mice after 9 weeks of MP administration. There was no difference in the body weight of the WT and Lep KO mice between the vehicle-treated and MP-treated groups, although the weight of the Lep KO mice was higher by 272.4% and 273.6% than that of the WT mice (Figure 2A). This change in body weight was completely reflected in their appearance (Figure 2B). Also, there was no significant alteration in the weight and gross morphology of the liver between the vehicle- and MP-treated WT and Lep KO mice (Figure 2C). Therefore, these results indicate that the MP administration does not cause body and liver weight gain in WT and Lep KO mice.

### 2.3. Effect of MPs on Lipid Metabolism in the Liver Tissue

Next, we investigated whether the oral administration of MPs could affect the lipid metabolism in the liver tissue. To achieve this, alterations in the fat accumulation, adipogenesis, lipogenesis, and lipolysis in the liver tissue of WT and Lep KO mice were analyzed after an oral administration of MPs for 9 weeks. The histopathological analyses used to evaluate fat accumulation were analyzed in hematoxylin and eosin (H&E)-stained liver sections. The number of lipid droplets and the areas of steatosis and macrovascular steatosis were significantly decreased by 26.0–49.8% in the MP-treated Lep KO mice compared to the vehicle-treated Lep KO mice, without any difference in their non-alcoholic fatty liver disease (NAFLD) scores. However, in the WT mice, the MPs treatment resulted in an increase in the NAFLD score and steatosis area (604.6%) compared to the vehicle-treated group (Figure 3A,B). These results showed that the oral administration of MPs may affect fat accumulation in the liver tissues of both mice, though the pattern of fat accumulation may differ between the two types of mice.

The analyses of adipogenesis and lipogenesis revealed an overall similarity in the transcription levels of the genes encoding the adipogenic transcription factors peroxisome proliferator-activated receptor gamma [PPARγ] and CCAAT enhancer-binding protein alpha [C/EBPα], although there was a difference in their regulation between the WT and Lep KO mice. The levels of PPARγ and C/EBPα were increased by 74.0% and 72.0% in the MP-treated WT mice compared to the vehicle-treated WT mice, while they were decreased by 22.7% and 15.6% in the corresponding groups of Lep KO mice (Figure 4A,B). However, the transcription levels of two lipogenic genes (adipocyte protein 2 [aP2] and fatty acid synthase [FAS]) showed the reverse pattern in the MP-treated groups. The levels of FAS mRNA were increased by 62.0% and 45.8% in the MP-treated groups of the WT and Lep KO mice compared to their corresponding levels in the vehicle-treated groups (Figure 4C). In addition, the level of aP2 mRNA was decreased by 25.0% and 23.8% in the MP-treated groups of the WT and Lep KO mice compared to its corresponding level in the vehicle-treated groups (Figure 4D).

Meanwhile, in the lipolysis analyses, three key proteins showed a similar response pattern to the MPs treatment regardless of the mouse type. In comparison to the vehicle-treated group, the expression level of adipose triglyceride lipase (ATGL), as well as the levels of phosphorylated hormone-sensitive lipase (HSL) and perilipin, were remarkably decreased by 19.0–89.0% and 28.3–51.5% in the MP-treated group, although the extent of the decrease in the proteins was variable. Specifically, these decrease rates were higher in the WT mice than in the Lep KO mice (Figure 5). Therefore, the above results suggest that the disruption of lipid accumulation in the liver after the oral administration of MPs for 9 weeks could be closely linked to the observed alterations in lipogenesis, adipogenesis, and lipolysis through the dysregulation of the adipogenic transcription factors, the lipogenic genes, and the lipolytic proteins.

### 2.4. Effect of MPs on the Serum Lipid Profile and the Metabolism of Lipids in the Adipose Tissues

Hepatic fat accumulation can be controlled by four major routes, including the uptake of lipids from circulation, de novo lipogenesis (DNL), fatty acid oxidation (FAO), and lipid export through very-low-density lipoproteins (VLDL) [22]. To investigate whether the MP-induced disruption of lipid metabolism in the liver tissue was accompanied by a dysfunction of the metabolism in the extrahepatic adipose tissue, alterations to the lipid profiles, fat accumulation, lipogenesis, and lipolysis were analyzed in the serum and adipose tissues of WT and Lep KO mice orally administrated MPs for 9 weeks. First, the changes in the hepatic lipid metabolism affected the lipid profile, including TC, TG, high-density lipoprotein (HDL), and low-density lipoprotein (LDL), in serum. In the WT mice, the concentrations of TC, TG, and LDL were significantly increased by 22.9%, 87.2%, and 11.9% in MP-treated group compared to the vehicle-treated group, while that of HDL remained constant. In the Lep KO mice, the MP administration did not induce any significant alterations compared to the vehicle-treated group (Table 1).

Also, the analyses of the adipose tissues showed that their weights and adipocyte areas remained constant in the vehicle- and MP-treated groups of the WT and Lep KO mice, although their levels were higher in the Lep KO mice than in the WT mice (Figure 6). However, significant differences between the vehicle- and MP-treated groups were observed in the alterations in their adipogenesis, lipogenesis, and lipolysis. The transcription levels of the genes encoding two adipogenic transcription factors (PPARγ and C/EBPα) and the two lipogenic genes (aP2 and FAS) were remarkably decreased by 33.0–51.0% and 33.3–75.0% in the MP-treated group compared to the vehicle-treated group. These patterns were maintained consistently in the WT and Lep KO mice, although the overall expression levels were lower in the Lep KO mice than in the WT mice (Figure 7). However, an inverse pattern was detected in their lipolysis regulation. The expression level of ATGL, as well as the phosphorylation levels of HSL and perilipin, was significantly enhanced by 16.0–9045.0% and 109.3–443.7% in the MP-treated group compared to the vehicle-treated group, although the increase rate of each protein was varied (Figure 8). Finally, we analyzed whether the regulation of leptin, a hormone related to satiety and appetite inhibition secreted by the adipose tissue, was associated with lipid metabolism. The concentrations of leptin in the adipose tissue and serum were significantly increased by 147.0% and 22.0% in the MP-treated WT mice compared to the vehicle-treated WT mice. Leptin was not detected in the Lep KO mice due to their genetic deficiency (Figure 9A,B). However, the leptin increases caused by the MPs treatment did not affect the food intake or water consumption in the WT and Lep KO mice (Figure 9C,D). Therefore, the above results suggest that the disruption of hepatic lipid metabolism caused by MPs’ accumulation could be closely linked to the dysregulation of lipid profiles in the serum and the lipid metabolism and leptin secretion in the adipose tissue.

### 2.5. Effect of MPs on Glu Metabolism in the Liver Tissue

Next, we investigated the effects of the administration of MPs on the Glu metabolism of the liver tissue. To achieve this, the alterations in glycogenolysis, the Glu transporter type 4 (GLUT4)-5′ AMP-activated protein kinase (AMPK) signaling pathway, and insulin resistance were analyzed in the liver tissues of WT and Lep KO mice orally administered MPs for 9 weeks. First, the impact of the MPs on glycogenolysis was investigated by analyzing the levels of Glu and key enzymes in the liver. The overall pattern was reversed in the concentration of Glu between the serum and the liver tissue. The Glu concentration was decreased by 22.9% and 43.3% in the serum of the MP-treated groups regardless of the mouse type, but it increased by 42.4% and 28.7% in the liver tissue of the same groups (Figure 10A,B). Also, similar patterns were detected in the transcription levels of the genes encoding the key enzymes for glycogenolysis. In comparison to the vehicle-treated group, their levels were significantly increased, by 30.0–58.0% and 36.2–92.6%, in the MP-treated group regardless of the mouse type (Figure 10C–E). These results showed that the oral administration of MPs can stimulate glycogenolysis in the liver tissues of WT and Lep KO mice.

Second, the impact of MPs on the GLUT4-AMPK signaling pathway was investigated by analyzing the GLUT4 expression and AMPK phosphorylation of the mice. The alterations of these factors were well reflected in the changes in the concentration of Glu in their serum. The levels of GLUT4 protein and mRNA were lower by 58.0–74.0% and 64.1–68.2% in the MP-treated group than in the vehicle-treated group of both mouse types (Figure 11A,B). The phosphorylation level of AMPK, as an upstream mediator, was similarly decreased by 20.0% and 14.5% after the MPs’ administration (Figure 11C). Thus, these results indicate that the oral administration of MPs can suppress the GLUT4-AMPK signaling pathway in the liver tissues of WT and Lep KO mice.

Finally, the impact of MPs on the regulation of insulin resistance was investigated by evaluating the alterations in the expression of insulin receptors and their downstream signaling pathway. The protein expression and the transcription level of genes encoding the insulin receptors were increased by 13.0–43.0% and 22.7–39.3% in the MP-treated group compared to those in the vehicle-treated group (Figure 12A,B). Also, a similar pattern was detected in the phosphorylation of three key members, including the phosphorylation of the insulin receptor substrate (IRS)-1, phosphoinositide 3-kinase (PI3K), and protein kinase B (Akt), under the insulin receptor downstream signaling pathway, although the range of their increase rates was very wide (Figure 12C). The above data suggest that the oral administration of MPs for 9 weeks could be closely linked to alterations in insulin resistance. Taken together, these results indicate that the administration of MPs can be linked to the disruption of Glu metabolism in the liver tissue through the stimulation of glycogenolysis, the suppression of the GLUT4-AMPK signaling pathway, and an increase in the insulin resistance of the liver tissue of WT and Lep KO mice.

### 2.6. Effect of MPs on Amino Acid Metabolism in the Liver Tissue

To investigate whether the MP-induced disruptions of lipid and Glu regulation in the liver tissue are accompanied by alterations in amino acid metabolism, a pattern recognition analysis of the endogenous metabolites for amino acid in the liver tissues of WT and Lep KO mice administered MPs for 9 weeks was conducted using a multivariate analysis. The NMR spectra, in global profiling, showed clustering between the vehicle- and MP-treated WT mice, as well as between the vehicle- and MP-treated Lep KO mice (Figure 13A). Also, the clusters of the vehicle-treated group were remarkably shifted after the MPs treatment in both mouse types (Figure 13A). A total of 19 metabolites were detected using the Chenomx NMR Suite, which was applied for a targeted NMR spectral analysis; the partial least squares discriminant analysis (PLS-DA) score plots differed between the vehicle- and MP-treated groups for both mouse types. Among them, fifteen endogenous metabolites were selected for an investigation into the effects of MP treatments based on their variable importance in projection (VIP) scores. Specifically, seven metabolites including aspartate, lysine, S-adenosyl-L-homocysteine (SAH), taurine, glutathione (GSH), arginine, and serine showed a significant change in levels in both mouse types after the MPs treatment (Figure 13B,C). They were classified into two groups: an amino acid group containing aspartate, lysine, arginine, and serine, and a metabolite group containing SAH, taurine, and GSH. Furthermore, these endogenous metabolites that were changed in level in the MP-treated group were associated with the phenylalanine, tyrosine, and tryptophan biosynthesis pathways (Figure 14). Thus, these results suggest that the MP-induced disruption of lipid and Glu regulation in the liver tissue could be closely associated with alterations in the amino acid metabolism and, especially, the phenylalanine, tyrosine, and tryptophan biosynthesis pathways.

### 2.7. Verification of the MP-Induced Disruption of Hepatic Metabolism in the HepG2 Cells and MDI-Stimulated 3T3-L1 Adipocytes

Finally, the MP-induced disruptions of lipid and Glu metabolism observed in the liver tissues of the WT and Lep KO mice were further verified in their HepG2 cells and 3-isobutyl-1-methylxanthine, dexamethasone, and insulin (MDI)-stimulated 3T3-L1 adipocytes after the MPs treatment. In the HepG2 cells, MPs were successfully transported into the cytoplasm after 24 h (Figure 15A). The level of PPARγ mRNA was remarkably increased by 156.0% and 555.0% in the MP-treated groups, while an inverse pattern was observed in the transcription levels of the aP2 gene (Figure 15B,C). Also, the transcription levels of the genes encoding the two enzymes for glycogenolysis were significantly increased by 20.0–28.0%, and 40.0–410.0% after the MPs treatment (Figure 15D,E). In MDI-stimulated 3T3-L1 adipocytes, fat accumulation was significantly decreased after the MPs treatment for 24 h (Figure 16B). In addition, a similar decrease was detected in the transcription levels of PPARγ and aP2 for the regulation of adipogenesis and lipogenesis (Figure 16C). However, in comparison to the vehicle-treated group, the levels of phosphorylated HSL and perilipin for lipolysis were increased only in the HiMP-treated group (Figure 16D). Thus, these results from the liver and fat cells provide additional evidence that the MP administration may be linked to the disruption of lipid and Glu metabolism in the liver tissues of WT and Lep KO mice.

## 3. Discussion

The liver is a critical organ in the human body that is responsible for several functions, including metabolism and the clearance of exogenous substances such as MPs [23]. Also, this organ has an important role in preserving and regulating the levels of lipids, Glu, and amino acids in the body [24]. However, MPs of various sizes have been reported to have significant impacts on the metabolism of these three nutrients in the liver, which may lead to the dysfunction of multiple organs, including the intestine, adipose tissue, and muscle. But, in contrast, other earlier studies have also suggested that only small sizes of MPs may not adversely affect liver function. Given the conflicting data, it was necessary to conduct a detailed investigation into the effects of small-sized MPs on liver function. Therefore, in our study, we analyzed the impact of 500 nm MPs on lipid, Glu, and amino acid metabolism in the livers of WT and Lep KO mice. The results of this study may provide an important clue to the systemic effects of small-sized MPs in the mammalian body, as well as additional evidence regarding the molecular mechanisms involved in the impact of MPs on liver function in normal and leptin-deficient induced-obesity mice models.

Meanwhile, MPs are taken up by special phagocytes such as macrophages, monocytes, neutrophils, and dendritic cells through passive penetration or active endocytosis, although these mechanisms are still only partially understood [25,26,27]. Both clathrin-mediated and caveolin-mediated endocytosis mediated the internalization of MPs with a 20–200 nm size in epithelial cells [28], melanoma cells [29], and kidney cells [30]. Also, phagocytosis/micropinocytosis involves MPs less than 1 um in size during uptake the process in Caco-2 cells and macrophages [28,31]. Several types of MPs can be internalized by epithelial cells through passive penetration [25,32,33]. Furthermore, MPs of various sizes (50, 100, and 1000 nm) were internalized in liver cell lines (HepG2 cells) after 4 h under a low-concentration treatment (10 μg/mL) [34]. A similar internalization pattern was observed in HepG2 cells exposed to 1 μm MPs after 24 h [35]. In the present study, PS MPs 500 nm in size were successfully internalized to HepG2 cells 24 h after treatment at 0.00125 wt% or 0.0025 wt% to investigate the effects of MPs on hepatic functions. The results of the present study are very similar to previous studies showing the successful uptake of MPs of various sizes in the liver cells. In addition, our results provided additional evidence of the significant internalization of different-sized MPs into hepatocytes. However, our study had some limitations, in that it did not measure the distributions of the administered MPs within exact locations such as the hepatocytes, Kupffer cells, and sinusoids of liver tissue. Moreover, the lack of clear analyses of MPs’ kinetics, including their absorption, distribution, excretion in WT and Lep KO mice, should be considered as a drawback of our study, although this is due to the limitations of analytical technology.

The homeostasis of lipids, including fatty acids and TG, is tightly controlled in the liver through the precise regulation of biochemical, signaling, and cellular pathways [36]. But these constancies are disrupted by various external factors. Among them, an excess accumulation of TG within the liver is evident in NAFLD, the most common chronic liver disease, which is also associated with other disease conditions, including obesity, insulin resistance, dyslipidemia, and type 2 diabetes [37]. Similar disruptions of lipid homeostasis in the liver were also detected after treatments with MPs of various particle sizes and types. The number of lipid droplets, lipid accumulation, and TG levels were significantly increased in ICR mice and their offspring after MPs treatments with a particle size of less than 50 μm [13,15,16]. A similar enhancement in NAFLD activity score and lipid accumulation was detected in the liver of an HFD-induced obesity mouse model after an MPs treatment [19,20]. However, mice treated with MPs (0.5 and 50 μm) for 5 weeks showed opposite responses, which include a decrease in fat weight and the suppression of lipogenesis and mRNA levels for the enzymes involved in TG synthesis in the liver tissue [14]. The impairment of lipid deposition and alterations in the hepatic lipid species and transcriptional profiles in the lipid metabolic pathways were detected in C57BL/6 mice treated with MPs for 8 weeks [18]. In the current study, significant disruptions in the number of lipid droplets, NAFLD score, steatosis area, adipogenesis, lipogenesis, and lipolysis were detected in the liver tissues of WT and Lep KO mice treated with small-sized MPs for 9 weeks, without alterations in their body and liver weight. Treatment with MPs with a size of 0.5 μm induced several responses related to the stimulation of fat accumulation in the WT mice, while some responses related to the suppression of fat accumulation were detected in the Lep KO mice after the MPs treatment. These findings in WT mice are consistent with those from previous studies in mice co-treated with HFD and MPs, but Lep KO mice tended to have contrary responses. The difference in the MP-induced response between the Lep KO mice and the HFD-treated mice was attributed to the diversity of mechanisms that induce obesity, because some genetic and environmental factors are considered major causes of these diseases. The results of the present study provide additional scientific evidence that a treatment of small-sized MPs may be closely linked to the significant disruption of the homeostasis of lipids in the liver of WT and Lep KO mice. Moreover, the results of the present study provide a clue about the potential molecular mechanisms involved in the impacts of MPs on hepatic lipid accumulation. Generally, these mechanisms consist of an uptake of circulating lipids, de novo lipogenesis (DNL), fatty acid oxidation (FAO), and an exportation of lipids [22,38]. Among them, de novo lipogenesis converts acetyl-CoA (originating from excess carbohydrates) to new fatty acids, while fatty acid oxidation controlled by PPARα could be considered the main mechanism of the impact of MPs, although further research is needed [22].

The lipid profile, including the TG, TC, HDL, and LDL in serum, is well reflected in the function of the liver, because this organ plays an essential role in lipid’s metabolism, synthesis, storage, and transportation [39]. Therefore, lipid profiles are considered indicators used to analyze the impact of MPs on the lipid metabolism in the liver. However, the findings of earlier studies show diverse impacts of small-sized MPs treatments on lipid profiles. The concentrations of TC and TG in serum decreased in ICR mice treated with polystyrene (PS) MPs (5 and 20 μm) for 28 days or 5 weeks [13,14]. In other studies, an opposite response in the serum lipid profile has also been reported. C57BL/6 mice co-treated with HFD and MPs of 0.45–0.53 μm for 4 weeks showed an increase in their TG and LDL concentrations [20], while no significant alterations were observed in the TG and TC concentrations in the corresponding group of mice treated with MPs of 1 μm for 8 weeks [18]. In addition, a positive correlation between TG concentrations and MPs treatments was detected in mice after a treatment with MPs of particle sizes of 5 or 20 μm [16]. In the present study, a significant increase in the concentrations of TC, TG, and LDL was detected only in the WT mice, and not in the Lep KO mice, after their treatment with MPs of 500 nm. Our findings using 500 nm sized particles were consistent with only a few previous studies that have analyzed the lipid profile in MP (5 or 20 μm)-treated mice for 28 days and C57BL/6 mice co-treated with HFD + MPs of 0.45–0.53 μm for 4 weeks [16,20]. The results of the present study provide additional scientific evidence of the close relationship between a treatment with small-sized MPs and an increase in lipid profile. However, further research will be needed to explain the clear difference between them.

After the secretion of leptin from the adipose tissue, it plays an important role by providing signals that regulate food intake, energy expenditure, and neuroendocrine function [40]. This hormone participates in the regulation of lipid metabolism in the peripheral tissues [41]. Also, the production and secretion of leptin are affected by a wide range of factors, including fasting, energy restriction, feeding, surgical stress, and several hormones [42,43,44,45]. However, the impact of MPs of various sizes on the regulation of leptin has never been studied until now. In this research, we initially examined the impact of MPs on leptin concentrations in the adipose tissue and serum. The concentration of this hormone was remarkably increased in WT mice following their treatment with small-sized MPs, although its increase did not affect the dietary intake and water consumption of the WT mice. Thus, the results of the present study provide the first scientific evidence that small-sized-MP administrations for 9 weeks may be closely linked to the upregulation of the leptin concentration in the adipose tissues and serum of C57BL/6 mice. However, additional studies are needed to prove the molecular mechanism of the impact of small-sized MPs on the regulation of leptin’s expression and secretion in the adipose tissue of mice.

Meanwhile, the impact of MPs on Glu metabolism has been studied in a very limited manner in mice thus far, even though the liver plays a major role in the production, storage, and transportation of Glu [46]. Blood Glu concentrations were increased in mice treated with 5 mg/kg or 15 mg/kg of MPs of 0.45–0.53 μm for 21 days, as well as in carboxyl-modified fluorescent PS MPs (80 nm)-treated C57BL/6 mice fed an HFD for 4 weeks [17,20]. Also, an impairment of Glu tolerance was detected in C57BL/6 mice treated with 0.1 μm-sized MPs (100 μg/L and 1000 μg/L) for 8 weeks [18]. In the present study, treatment with 500 nm-sized MPs induced an increase in hepatic Glu concentration and the transcription of the enzymes associated with glycogenolysis, as well as the suppression of GLUT4 expression and AMPK phosphorylation in the liver tissues of MP-treated WT and Lep KO mice. Our findings are consistent with previous studies with respect to the increase in blood Glu concentrations after MPs treatments, although the findings of the current study also include additional mechanisms. Thus, the results of the present study provide additional evidence that treatment with small-sized MPs may be closely associated with the upregulation of blood Glu concentrations.

In the present study, the expression levels of the insulin receptor and the phosphorylation levels of key mediators in its downstream signaling pathway were analyzed in the liver tissue of MP-treated mice to investigate the impact of MPs on insulin resistance. These levels of the insulin receptor and IRS-1, PI3K, and Akt were enhanced in both mouse types after MP administrations for 9 weeks. These findings of the present study are only partially consistent with the results of earlier studies, although the duration of administration is different. Increases in insulin levels were detected in Wistar rats treated with MPs (2.5, 5, and 10 mg/kg/day) for 45 days, but mice treated with MPs (5 and 15 mg/kg) for 21 days did not show any significant alteration in insulin secretion [17,47]. Therefore, our results provide additional evidence that the MP-induced disruption of the lipid and Glu metabolism in the liver is associated with the disruption of insulin resistance. In addition, the data from the present study provide novel scientific evidence that a disruption in insulin resistance induced by 45 days of MPs treatment remains constant with that of a treatment for 90 days.

Finally, we investigated whether the MP-induced disruption of the Glu and lipid metabolism may accompany alterations in amino acid metabolism, since endogenous metabolites are sensitive to various environmental factors, including diet, stress, xenobiotics, and lifestyle [48]. However, earlier studies have provided only limited information about the impact of MPs in this aspect. Significant alterations in metabolite profiles were detected in the serum of MP (5 μm and 20 μm)-exposed mice. Among a total of 35 differential metabolites (DMs), 37 DMs including 8 DMs (energy metabolism), 4 DMs (lipid metabolism), 3 DMs (response to oxidative stress), and 4 DMs (response to neurotoxin) were detected in MP-treated mice [13]. Also, the F1 offspring born to pregnant ICR mice after an MPs treatment showed different responses according to gender. Several amino acids, including arginine, methionine, ornithine, phenylalanine, proline, tyrosine, and propionyl carnitine, were remarkably decreased in the serum of F1 female mice, while free carnitine was increased in the same mice [15]. In the serum of MP-treated F1 male offspring, some metabolites such as leucine/isoleucine/hydroxyproline, acetylcarnitine, and propionylcarnitine decreased, but others, including alanine, citrulline, proline, and free carnitine, increased [15]. In the present study, seven metabolites, including aspartate, lysine, SAH, taurine, GSH, arginine, and serine, that changed after an administration of MPs for 9 weeks were identified in the liver tissue of WT and Lep KO mice. Among them, decreases in arginine levels were observed in earlier studies and the current study, although the samples analyzed were different. Furthermore, the correlation between an MPs treatment and alterations in arginine levels has only been reported in a few studies. A significant disruption in arginine-related pathways was observed in *Microcystis aeruginosa*, a common freshwater microalga, after an MPs (0.1, 1, 10, and 100 μm) exposure for 96 h [49]. Therefore, these findings provide novel evidence that arginine-related pathways could be considered an important target while assessing the impact of MPs on liver metabolism.

## 4. Materials and Methods

### 4.1. Preparation of MPs for Treatment

The dispersed MPs suspension for use in the experiment was prepared and characterized as described in earlier studies [50]. The MPs suspension comprised PS materials, with a mean diameter size of 0.4–0.6 μm and a density of 1.04–1.06 g/cm^3^ at a concentration of 25 mg/mL, in aqueous suspension (Sigma-Aldrich Co., St. Louis, MO, USA). Also, the MPs consisted of circular-shaped particles with a specific number–distribution size (607.23 ± 8.27 d.nm) and zeta potential (−35.52 ± 1.18 mV) (Figure 17). The size of the MPs used in both mice was determined based on the previous reports on MPs as cause of constipation [51], autophagy [52], and inflammation [53].

### 4.2. Production and Identification of Lep KO Mice

Lep KO mice were produced and identified as described in earlier studies [54,55]. Seven-week-old leptin heterogynous (HT) mice of the C57BL/6J strain were supplied by the Department of Laboratory Animal Resources (Laboratory Animals Resources Bank) at the National Institute of Food and Drug Safety Evaluation (NIFDS, Chungju, Republic of Korea). To generate the WT and Lep KO mice used in this study, 8-week-old male and female HT mice were mated at a ratio of 1:2. Subsequently, the genotypes of their offspring were analyzed using PCR with genomic DNA extracted from the tails of 1-week-old founder mice. Exon 2 of the leptin gene was specifically amplified using specific primer sets (Appendix A). PCR was performed with 30 cycles of amplification on a T100™ Thermal Cycler (Bio-Rad Laboratories Inc., Hercules, CA, USA) under the following conditions: 30 s, 95 °C; 30 s, 60 °C; 30 s, 72 °C. Finally, the PCR products were separated in 2% agarose gel with ethidium bromide (EtBr) and identified as 219 bp and 140 bp under a UV visualizer (Atto, Tokyo, Japan).

### 4.3. Design of the Animal Experiment

The animal experiment protocol was reviewed and approved by the Pusan National University-Institutional Animal Care and Use Committee (PNU-IACUC) (Approval Number PNU-2022-0191, PNU-2023-0350). All experiments related to mice were performed at the PNU-Laboratory Animal Resources Center, accredited by the MFDS (Accredited Unit Number-000231) and the Association for the Assessment and Accreditation of Laboratory Animal Care (AAALAC) International (Accredited Unit Number; 001525). All the mice were provided with a standard diet (Samtako Bio Korea Co., Osan, Republic of Korea) comprising nitrogen-free extracts (50.5%), crude protein (22.5%), moisture (11.0%), fat (5.0%), fiber (4.5%), and ash (6.5%). The mice were given access to filtered tap water ad libitum. The mice were maintained in a strict light–dark cycle (08:00 h to 20:00 h), consistent temperature (22 ± 2 °C), and humidity (50 ± 10%), under a specific pathogen-free (SPF) state.

Seven-week-old WT (*n* = 12) and Lep KO mice (*n* = 12) were further divided into two experimental subgroups each: the vehicle-treated WT mice (*n* = 6), MP-treated WT mice (*n* = 6), vehicle-treated Lep KO mice (*n* = 6), and MP-treated Lep KO mice (*n* = 6). The vehicle-treated groups were orally administered the same volume of 1 × phosphate-buffered saline (PBS) solution, while the MP-treated groups were orally administered the dispersed MPs suspension at a concentration of 100 μg/mL, once daily (0.5 mL/day) three times a week for 9 weeks. The dosage of MPs administered to WT and Lep KO mice was determined based on an earlier study that prove the oral administration of dispersed MP suspensions (10 μg/L, 50 μg/L, and 100 μg/L) once daily (0.5 mL/day) caused chronic constipation [51]. At 24 h after the last administration, the WT and Lep KO mice were euthanized using an appropriate chamber equipped with a gas regulator and CO_2_ gas with a minimum purity of 99.0%. Finally, the liver and adipose fat tissues were collected from the mice and split into subgroups for histopathological analyses and molecular assays.

### 4.4. Measurement of Body and Weight of Organs

The body weight of each mouse in the experimental groups was analyzed using an electronic balance (Mettler Toledo, Greifensee, Switzerland). Also, the weights of the liver and adipose tissues (epididymal and retroperitoneal fats) collected from the sacrificed WT and Lep KO mice were determined using the same method used to ascertain their body weight. Images of the two organs were captured with a digital camera (Samsung, Seoul, Republic of Korea).

### 4.5. Measurement of Food Intake and Water Consumption

The remaining quantity of feed and water was measured every morning using an electronic balance (Mettler Toledo) and a mass cylinder, during the experimental period. Finally, the value of the food intake and water consumption of each mouse was calculated as follows:Food intake (Water consumption) = Initial supply − Remaining amount

### 4.6. Histopathological Analysis

The liver and adipose tissues collected from the mice belonging to each experimental group were fixed in 10% formalin solution over 48 h. Subsequently, these tissues were embedded in paraffin wax and cut into 4 µm-thick slices. The paraffin sections were deparaffinized with xylene, hydrated with ethanol (100 to 70%), and stained with H&E solution (Sigma-Aldrich Co.). Each section on the slide was dehydrated with ethanol (70 to 100%) and cleared with xylene. Alterations of the histopathological features were microscopically examined at 400× magnification under a light microscope (Leica Microsystems, Wetzlar, Germany).

The lipid accumulation in the liver and adipose tissues was measured by counting the number of lipid droplets using the Leica Application Suite (Leica Microsystems) and analyzing the areas of the adipocytes using the Image J 1.52a program (National Institutes of Health [NIH], Bethesda, ML, USA). Also, the H&E-stained sections of the liver tissue were scored for NAFLD based on the histopathological features of steatosis (0–3), inflammation (0–4), and ballooning (0–2), as described in an earlier study (Appendix A) [21].

### 4.7. Serum Biochemical Analysis

Whole blood was collected from each mouse using a 1 mL syringe with a 27G × 1/2″ needle. After incubation in a serum-separating tube (BD Containers, Franklin Lakes, NJ, USA) for 10 min, the total serum was obtained by centrifugation at 1500× *g* for 15 min. The levels of various components including Glu, TC, TG, HDL, and LDL were analyzed using a BS-120 Automatic Chemical Analyzer (Mindray, Shenzhen, China).

### 4.8. RT-qPCR Analyses

The total RNA was purified from the frozen tissues of the mice, as well as their HepG2 and 3T3-L1 cells, using the Tri-RNA Reagent (Favorgen Biotech, Ping-Tung, Taiwan) based on the guidelines provided by the manufacturer. After homogenizing the tissues (50 mg of liver and 100 mg of adipose tissue) and cells (1 × 10^5^ of 3T3-L1 cells and 2 × 10^5^ of HepG2 cells) in the Tri-RNA Reagent, total RNAs were extracted by centrifugation at 15,000× *g* rpm for 5 min, and their concentration was measured using a nano-300 micro-spectrophotometer (Allsheng Instruments Co. Ltd., Hangzhou, China). Total complementary DNA (cDNA) was synthesized from the mRNA template using Superscript II reverse transcriptase (Thermo Scientific, Waltham, MA, USA). Each gene was amplified from a mixture containing cDNA templates, 2 × Power SYBR Green (Toyobo Life Science, Osaka, Japan), and specific primers (Appendix A). The PCR progressed for 40 cycles consisting of the following steps: denaturation at 95 °C for 15 s followed by annealing and extension at 70 °C for 60 s. The fluorescence intensities of each sample were analyzed at the end of the extension phase of each cycle. Finally, the cycle quantification value (Cq) was explained as described in the 2^−ΔΔCT^ method [56].

### 4.9. Western Blot Assay

The total tissue and cell homogenates were extracted from the tissues (50 mg for liver and 100 mg for adipose tissues) and 3T3-L1 cells (1 × 10^5^) into Pro-Prep Protein Extraction Solution (iNtRON Biotechnology, Seongnam, Republic of Korea) using a Polytron PT-MR 3100 D Homogenizer (Kinematica AG, Lusern, Switzerland), and the protein concentration of each sample was determined using a SMART™ BCA Protein Assay Kit (Thermo Scientific). Homogenated protein samples (30 μg) were loaded into the wells and separated using 8–10% SDS-PAGE for 2 h. Subsequently, the resolved proteins were transferred to nitrocellulose membranes for 2 h 30 min at 40 V, after which the membranes were incubated at 4 °C for 12 h with specific primary antibodies (Appendix A). Subsequently, the membranes with primary antibodies were washed with washing buffer (137 mM NaCl, 2.7 mM KCl, 10 mM sodium hydrogen phosphate [Na_2_HPO_4_], and 0.05% Tween 20) to remove the unconjugated primary antibody and incubated with 1:10,000 diluted horseradish peroxidase (HRP)-conjugated goat anti-rabbit Immunoglobulin (Ig) G (Invitrogen, Waltham, MA, USA) at room temperature (RT) for 1 h. Finally, the antibody-conjugated membranes were developed using the EZ-Western Lumi Femto Kit (Dogen Bio, Seoul, Republic of Korea). The fluorescence of the proteins was detected using the FluorChem^®^FC2 imaging system (Alpha Innotech Co., San Leandro, CA, USA). Finally, the band density was measured using Evolution Capt software version 18.02 (Vilber Lourmat Deutschland GmbH, Eberhardzell, Germany).

### 4.10. Measurement of Leptin Concentrations

The concentrations of leptin in the serum and adipose tissues were quantified using enzyme-linked immunosorbent assay (ELISA) kits (Invitrogen) according to the guidance provided by the manufacturer. The frozen adipose tissues (100 mg) were homogenized in radioimmunoprecipitation assay (RIPA) Lysis Buffer (Merck Millipore, Darmstadt, Germany) using a Polytron PT-MR 3100 D Homogenizer (Kinematica AG). Subsequently, their supernatants were collected for the ELISA analysis. The samples, including serum and tissue homogenates and standards, were incubated in a 37 °C incubator for 2 h on plates coated with leptin antibody. After washing the wells four times, biotin conjugate was added to each well, and these mixtures were incubated at RT for 1 h. This was followed by the addition of the HRP conjugate and incubation at RT for 30 min, after which the reaction was halted by the addition of a stop solution. The absorbance of each well was measured using the VersaMax™ Plate Reader (Molecular Devices, San Jose, CA, USA).

### 4.11. Measurement of Glu Concentrations

Glu is oxidized by the catalytic action of Glu oxidase (GOD) to produce hydrogen peroxide and, in the presence of Glu peroxidase (POD), hydrogen peroxide oxidizes 4-aminoantipyrine, along with ρ-hydroxybenzoic acid sodium, to form quinonimine [57]. The Glu concentrations were measured based on the GOD-POD method. Briefly, the total protein lysates were extracted from the liver tissue (40 mg) using a Polytron PT-MR 3100 D Homogenizer (Kinematica AG) in a 1 × PBS solution. After mixing the lysate and GOD-POD, their absorbance was measured using a BS-120 Automatic Chemical Analyzer (Mindray).

### 4.12. LC-MS Analysis of Metabolites

After homogenizing the liver with a four-fold volume of buffer (150 mM NaCl, pH 7.4), the obtained tissue lysate was mixed with a three-fold volume of methanol. The supernatant collected by centrifugation at 13,000× *g* rpm for 20 min at 4 °C was diluted two-fold with distilled water. Following the derivatization of the sample using o-phthaldialdehyde (OPA, Merck KGaA, Darmstadt, Germany)/2-mercaptoethanol (Merck KgaA), it was injected into a high-performance liquid chromatography (HPLC) system equipped with a Hector C18 column (3 µm × 4.6 mm × 100 mm; Rstech Corp, Cheongju, Republic of Korea). The peaks for each sample were detected using a fluorescence detector (excitation at 338 nm and emission at 425 nm; Thermo Fisher Scientific).

For measuring the levels of S-adenosylmethionine (SAM) and SAH, the liver samples were homogenized in 6% perchloric acid (Merk KgaA). Their supernatant was injected into an HPLC system equipped with a Hector C18 column (5 µm × 4.6 mm × 250 mm; Rstech Corp). The peaks of SAM and SAH were detected at 254 nm using an ultraviolet (UV) detector (Thermo Fisher Scientific).

For determining the levels of GSH and taurine, the liver was homogenized in 1 M perchloric acid, which contained either 2 mM ethylenediamine tetraacetic acid (EDTA) or methanol. Thereafter, it was centrifuged at 13,000× *g* rpm for 20 min at 4 °C and the supernatant was used to determine the GSH levels. Taurine was derivatized with OPA reagent (Merck KGaA)/2-mercaptoethanol (Merck KGaA) and quantified using an HPLC system equipped with a fluorescence detector (Thermo Scientific) after separation using a Hector T-C18 column (3 µm × 4.6 mm × 100 mm; Chiral Technology Korea, Daejeon, Republic of Korea).

The data were acquired using the Chenomx NMR Suite (v7.1) program (Chenomx Inc., Edmonton, AB, Canada) by quantifying the NMR spectra and probabilistic quotient normalization (PQN) was applied for normalization. The PLS-DA results were presented through scatter scores and a VIP score. The heatmap hierarchical clustering, revealing the metabolic similarity among the samples, was examined using the Pearson distance formula and Ward’s clustering algorithm. Biochemical pathway analysis was performed utilizing algorithms incorporating a global test and relative-betweenness centrality for a topology-based pathway enrichment analysis.

### 4.13. Cell Culture and Experiment

HepG2 and 3T3-L1 cells were obtained from the American Type Culture Collection (ATCC, Manassas, VA, USA). The 3T3-L1 cells were cultured in Dulbecco’s Modified Eagle Medium (DMEM, Welgene, Gyeongsan, Republic of Korea) containing 10% bovine calf serum (BCS, Welgene) and Penicillin/Streptomycin (Pen/Strep, Thermo Scientific), while HepG2 cells were incubated with Minimum Essential Medium, Eagle (MEM, Welgene) containing 10% FBS (Welgene) and Pen/Strep (Thermo Fisher Scientific). They were incubated at 37 °C with 5% CO_2_ and 95% fresh air.

The differentiation of 3T3-L1 cells was initiated according to the procedure outlined in an earlier study [58]. When the 3T3-L1 cells were confluent up to 80–90% in the culture plate, the culture media was removed using 10% BCS. After that, the differentiation (MDI) medium, including 3-isobutyl-1-methylxanthine (0.5 mM, Sigma-Aldrich Co.), dexamethasone (1 μM, Sigma-Aldrich Co.), and insulin (5 μg/mL, Sigma-Aldrich Co.) in DMEM supplemented with 10% FBS, was added (differentiation day 0). After incubation for 2 days, the MDI medium was replaced by normal medium containing 5 μg/mL of insulin for 48 h (differentiation day 2). Finally, the cells were maintained in normal medium with 10% FBS for 48 h (differentiation day 4).

To analyze the expression levels of proteins and genes, cells of each type were classified into three groups, namely the vehicle-treated cells, low-concentration-MPs-treated cells (LoMPs-treated group), and high-concentration-MPs-treated cells (HiMPs-treated group). The cells in the vehicle group were treated with the same volume of 1 × PBS solution, while the LoMPs- and HiMPs-treated groups in the HepG2 and 3T3-L1 cells were treated with 0.00125 wt% or 0.0025 wt% (HepG2 cells), respectively, as well as 0.005 wt% or 0.01 wt% (3T3-L1 cells), respectively, of a dispersed MPs suspension for 24 h. The morphology of the MP-treated cells was observed at 100×, 200×, and 400× magnification using a light microscope (Leica Microsystems). After centrifugation of the cell suspension, the total proteins and RNAs were extracted from the harvested cells, and their transcription levels were measured as described for the tissue samples.

### 4.14. Statistical Analysis

The statistical analysis of each experimental group was conducted using the Unpaired Two-Sample *t*-test and the One-way Analysis of Variance (ANOVA), followed by a Tukey multiple comparisons test after assessing normality using the Shapiro–Wilk test in SPSS 27.0 (IBM Corp, Armonk, NY, USA). The MetaboAnalyst 4.0 software (Edmonton, AB, Canada) was used for analyzing the metabolite data. All values are expressed as means ± SD, and a *p*-value (*p* < 0.05) was considered statistically significant.

## 5. Conclusions

Taken together, the current study investigated the impact of MPs on lipid, Glu, and amino acid metabolism under normal and obese conditions. These results provide novel evidence that MPs’ administration for 9 weeks is closely correlated with disruptions of hepatic lipid metabolism, including lipid accumulation, lipogenesis, and lipolysis, as well as hepatic Glu metabolism including glycogenolysis, the GLUT4-AMPK signaling pathway, and insulin resistance (Figure 18). Also, the above data were accompanied by the disruption of amino acids in the liver tissue, and lipogenesis, lipolysis, and leptin production in the adipose tissue. Therefore, we conclude that our findings establish that MPs can be considered one of the novel causes of the disruption of liver metabolism. However, our study has some limitations, which are as follows: the study did not analyze the mechanism of action of MPs on inflammation, glucotoxicity, and glucagon regulation, and it does not provide any results for a direct comparison between the diet-induced obesity model and the gene-deficient-induced obesity model.

## Figures and Tables

**Figure 1 ijms-25-04964-f001:**
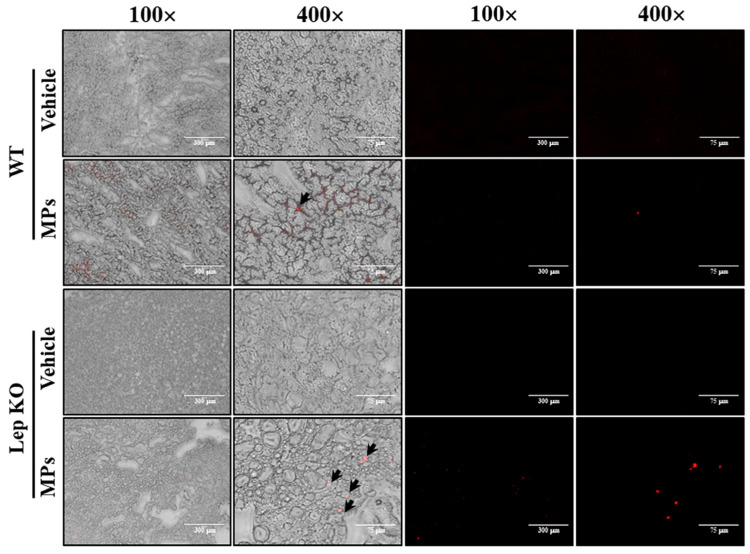
Accumulation of MPs in liver tissue. The red fluorescence of MPs was detected under a fluorescence microscope at 100× and 400× magnification. Arrows indicate MPs. Abbreviations: WT; wild type, Lep KO; leptin knockout, MPs; microplastics.

**Figure 2 ijms-25-04964-f002:**
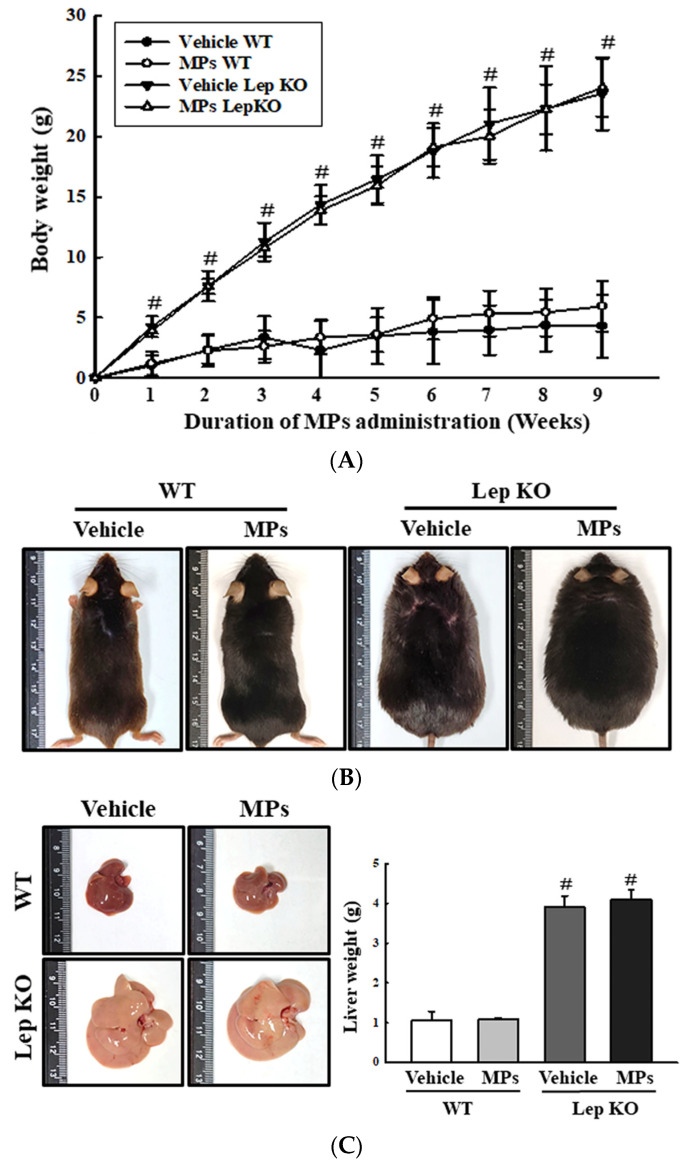
Body weight and weight of the liver. (**A**) Changes in the body weight of mice after 9 weeks of MP administrations. (**B**) Body appearance of WT and Lep KO mice after MP administration for 9 weeks. Analysis of body weight and appearance were performed on five to six mice per group, and each mouse’s weight was analyzed twice. (**C**) Morphology and weight of the liver. The preparation of liver samples was performed on five to six mice per group, and their weights were measured twice for each mouse. The data represent the mean ± SD. ^#^ indicates *p* < 0.05 compared to the WT mice. Abbreviation: WT; wild type, Lep KO; leptin knockout, MPs; microplastics.

**Figure 3 ijms-25-04964-f003:**
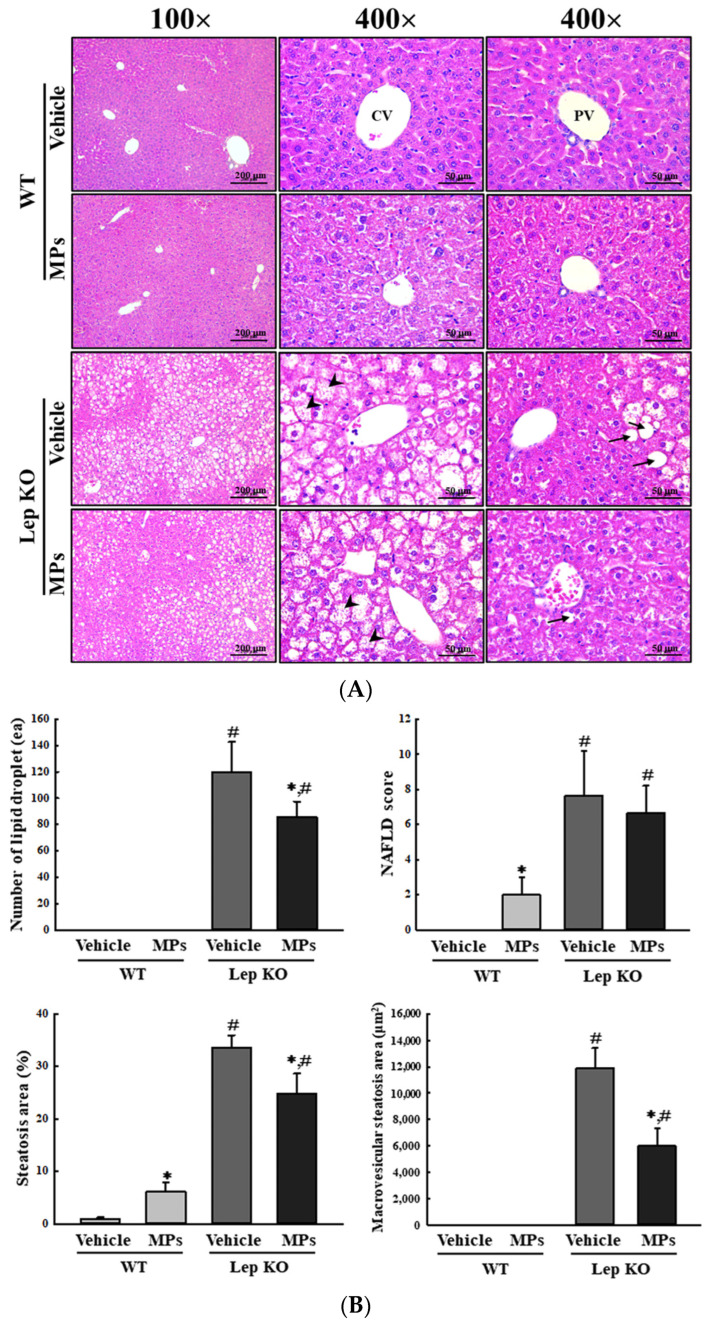
Alteration in the histopathological structure of the liver. (**A**) Histopathological structure of the liver. Pathological alterations were analyzed in H&E-stained liver sections at 100× and 400× magnification. Arrow heads indicate microvesicular steatosis and arrows indicate macrovesicular steatosis. (**B**) Assessment of the histopathological structure of the liver. The NAFLD scores in the H&E-stained liver sections were analyzed based on the degree of steatosis, inflammation, and ballooning as per an earlier study [21]. The H&E-stained sections of liver tissues were prepared from five to six mice per group, and their histopathological factors were analyzed twice per tissue. The data represent the means ± SD. * indicates *p* < 0.05 compared to the vehicle-treated group. ^#^ indicates *p* < 0.05 compared to the WT mice. Abbreviation: WT; wild type, Lep KO; leptin knockout, MPs; microplastics, H&E; hematoxylin and eosin, CV; central vein, PV; portal vein, NAFLD; non-alcoholic fatty liver disease.

**Figure 4 ijms-25-04964-f004:**
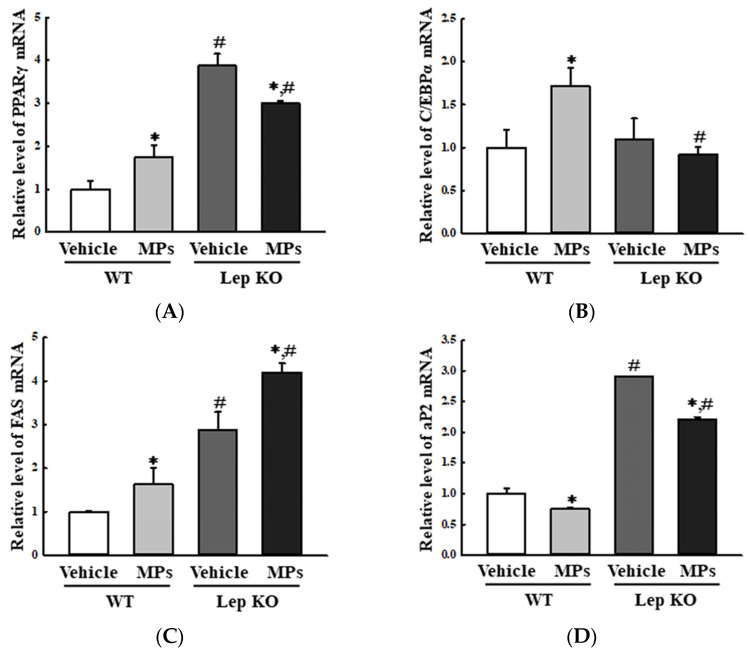
Transcription levels of genes encoding adipogenic and lipogenic factors. After the purification of the total RNA from the liver tissue, the mRNA levels of the genes encoding two adipogenic transcription factors (PPARγ (**A**) and C/EBPα (**B**)) and two lipogenic proteins (FAS (**C**) and aP2 (**D**)) were measured by RT-qPCR as described in the ‘Materials and Methods’ section. The preparation of the total RNA from the liver tissues was performed on five to six mice per group and the transcription levels of each gene were analyzed twice for each tissue. The data represent the means ± SD. * indicates *p* < 0.05 compared to the vehicle-treated group. ^#^ indicates *p* < 0.05 compared to the WT mice. Abbreviation: WT; wild type, Lep KO; leptin knockout, MPs; microplastics, RT-qPCR; reverse transcription-quantitative polymerase chain reaction, PPARγ; peroxisome-proliferator activator receptor γ, C/EBPα; CCAAT/enhancer binding protein α, FAS; fatty acid synthase, aP2; adipocyte protein 2.

**Figure 5 ijms-25-04964-f005:**
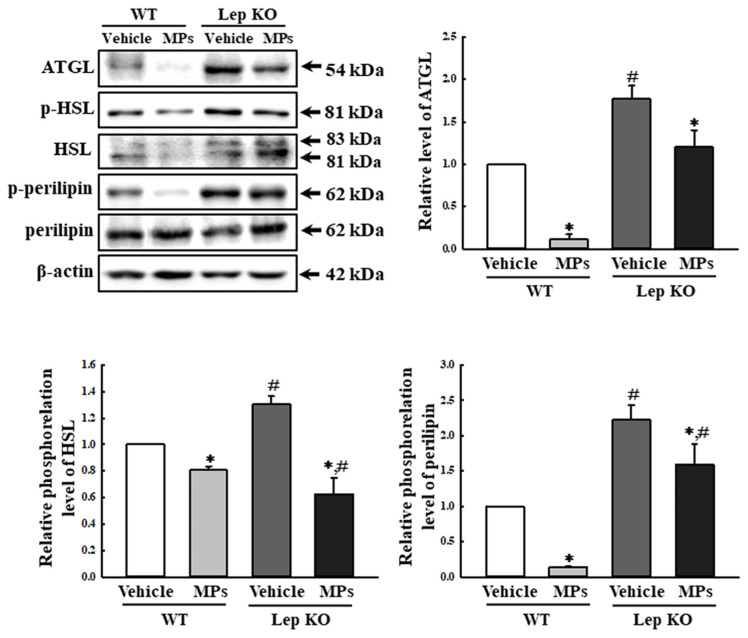
Expression and phosphorylation levels of lipolysis factors. After collection of the total proteins from the liver tissue, the expression levels of five lipolysis factors (ATGL, p-HSL, HSL, p-perilipin, and perilipin) were measured using a Western blot, as described in the ‘Materials and Methods’ section. The preparation of the total homogenates from the liver tissue was performed on five to six mice per group, and the expression levels of each protein were analyzed twice for each sample. The data represent the means ± SD. * indicates *p* < 0.05 compared to the vehicle-treated group. ^#^ indicates *p* < 0.05 compared to the WT mice. Abbreviation: WT; wild type, Lep KO; leptin knockout, MPs; microplastics, ATGL; adipose triglyceride lipase, HSL; hormone-sensitive lipase.

**Figure 6 ijms-25-04964-f006:**
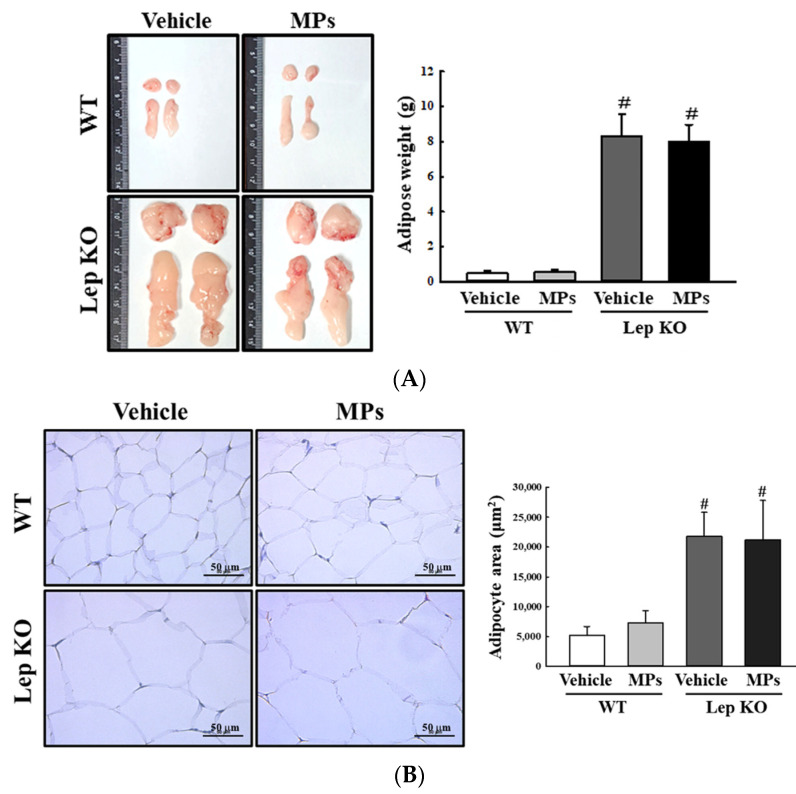
Weight, morphology, and histopathological structure of adipose tissues. (**A**) Morphology and weight of adipose tissues. The preparation of adipose tissue was performed using five to six mice per group, and their weights were measured twice for each sample. (**B**) Histopathological structure in the H&E-stained adipose sections. The H&E-stained sections of adipose tissue were prepared from five to six mice per group and the adipocyte area was measured twice for each stained tissue at 400× magnification. The data represent the means ± SD. ^#^ indicates *p* < 0.05 compared to the WT mice. Abbreviations: WT; wild type, Lep KO; leptin knockout, MPs; microplastics, H&E; hematoxylin and eosin.

**Figure 7 ijms-25-04964-f007:**
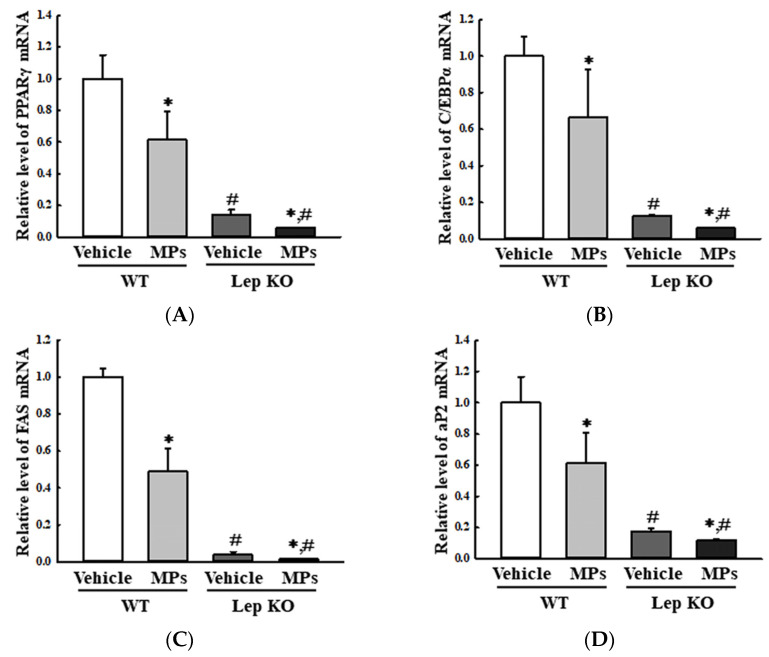
Transcription levels of the genes encoding adipogenic and lipogenic factors in the adipose tissue. After the collection of total RNA from the adipose tissue, the mRNA levels of PPARγ (**A**), C/EBPα (**B**), FAS (**C**) and aP2 (**D**) were measured by RT-qPCR. The preparation of the total RNA from the adipose tissue was performed on five to six mice per group, and the transcription levels of each gene were analyzed twice for each sample. The data represent the means ± SD. * indicates *p* < 0.05 compared to the vehicle-treated group. ^#^ indicates *p* < 0.05 compared to the WT mice. Abbreviations: WT; wild type, Lep KO; leptin knockout, MPs; microplastics, RT-qPCR; reverse transcription-quantitative polymerase chain reaction, PPARγ; peroxisome-proliferator activator receptor γ, C/EBPα; CCAAT/enhancer binding protein α, FAS; fatty acid synthase, aP2; adipocyte protein 2.

**Figure 8 ijms-25-04964-f008:**
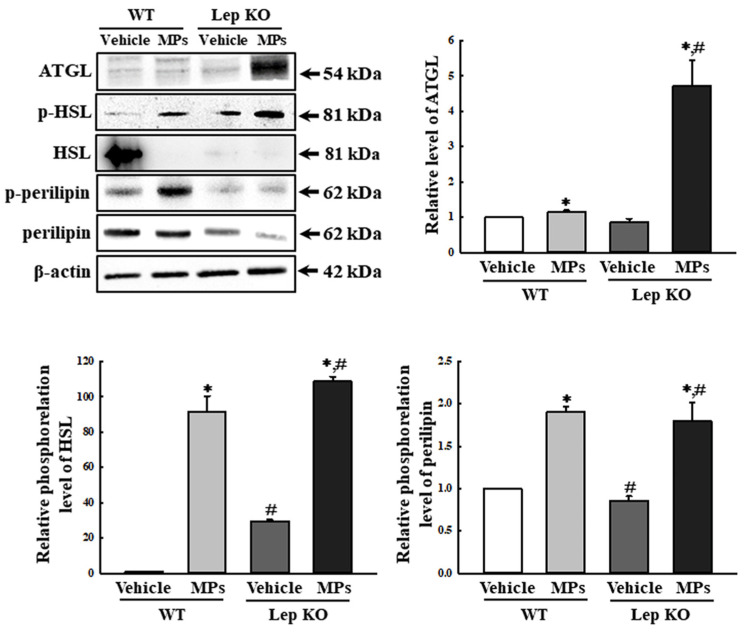
Expression and phosphorylation levels of the lipolysis factors in adipose tissues. After the collection of the total proteins from the adipose tissue, the expression levels of five lipolysis factors (ATGL, p-HSL, HSL, p-perilipin, and perilipin) were measured using a Western blot, as described in the ‘Materials and Methods’ section. The preparation of the total homogenates from the adipose tissue was performed on five to six mice per group, and the expression levels of each protein were analyzed twice for each sample. The data represent the means ± SD. * indicates *p* < 0.05 compared to the vehicle-treated group. ^#^ indicates *p* < 0.05 compared to the WT mice. Abbreviation: WT; wild type, Lep KO; leptin knockout, MPs; microplastics, ATGL; adipose triglyceride lipase, HSL; hormone-sensitive lipase.

**Figure 9 ijms-25-04964-f009:**
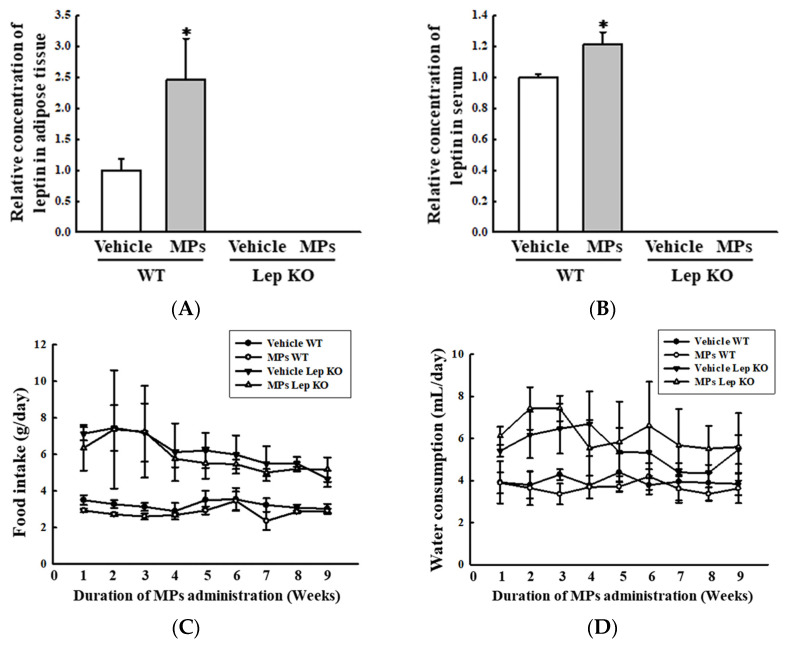
Level of the leptin hormone and feeding parameters. (**A**) Concentration of leptin in the adipose tissue. (**B**) Concentration of leptin in the serum. After collecting the serum and lysate of the adipose tissue, the level of this hormone was measured using an ELISA kit, which had a 20.0 pg/mL of analytical sensitivity and 63–4000 pg/mL of assay range. The preparation of serum and tissue lysate was performed on five to six mice per group; the concentrations of leptin were analyzed twice for each sample. (**C**) Level of food intake. (**D**) Level of water consumption. The measurements of food intake and water consumption were taken from five to six mice per group; these factors were analyzed twice for each mouse. The data represent the means ± SD. * indicates *p* < 0.05 compared to the vehicle-treated group. Abbreviations: WT; wild type, Lep KO; leptin knockout, MPs; microplastics, ELISA; enzyme-linked immunosorbent assay.

**Figure 10 ijms-25-04964-f010:**
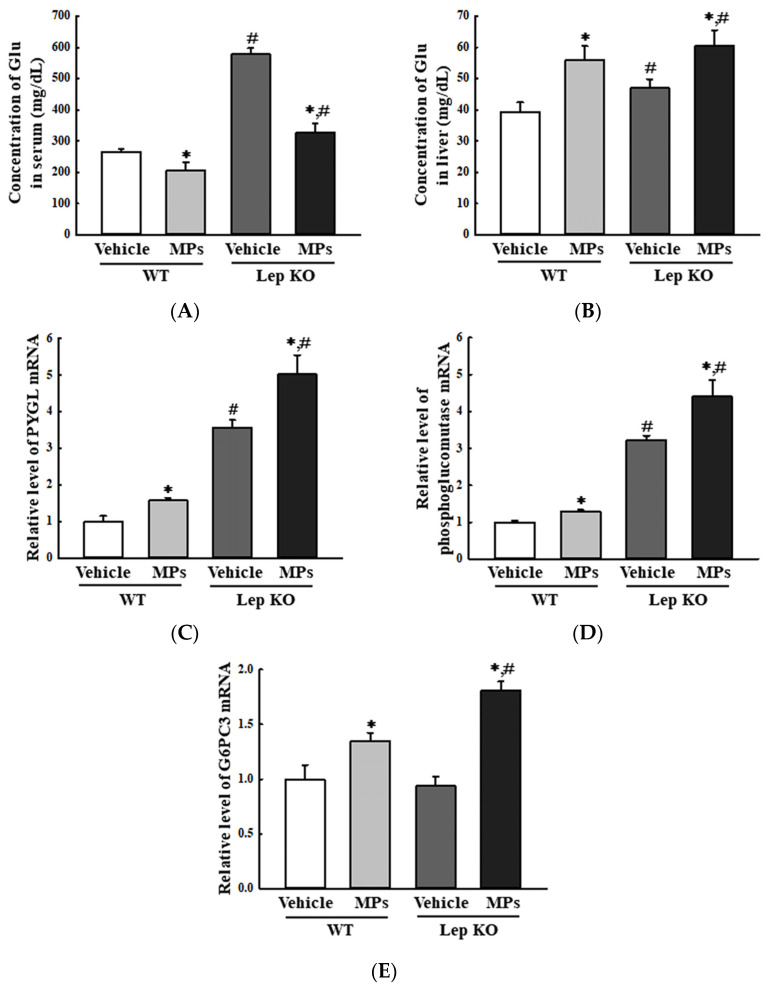
Level of the key regulators in the glycogenolysis and concentration of Glu. (**A**) Concentration of Glu in the serum. (**B**) Concentration of Glu in the liver tissue. After homogenizing liver tissue, the Glu concentrations were analyzed using an automatic chemical analyzer. (**C**–**E**) Transcription level of the genes encoding three enzymes for glycogenolysis. After the collection of total RNA from the liver tissue, the mRNA levels of the genes of the three enzymes for glycogenolysis were analyzed by RT-qPCR. The preparation of total RNA from the liver tissue was performed on five to six mice per group, and the transcription levels of each gene were analyzed twice for each tissue. The data represent the means ± SD. * indicates *p* < 0.05 compared to the vehicle-treated group. ^#^ indicates *p* < 0.05 compared to the WT mice. Abbreviations: WT; wild type, Lep KO; leptin knockout, MPs; microplastics, RT-qPCR; reverse transcription-quantitative polymerase chain reaction, Glu; glucose, PYGL; glycogen phosphorylase L, G6PC3; Glu-6-phosphatase catalytic subunit 3.

**Figure 11 ijms-25-04964-f011:**
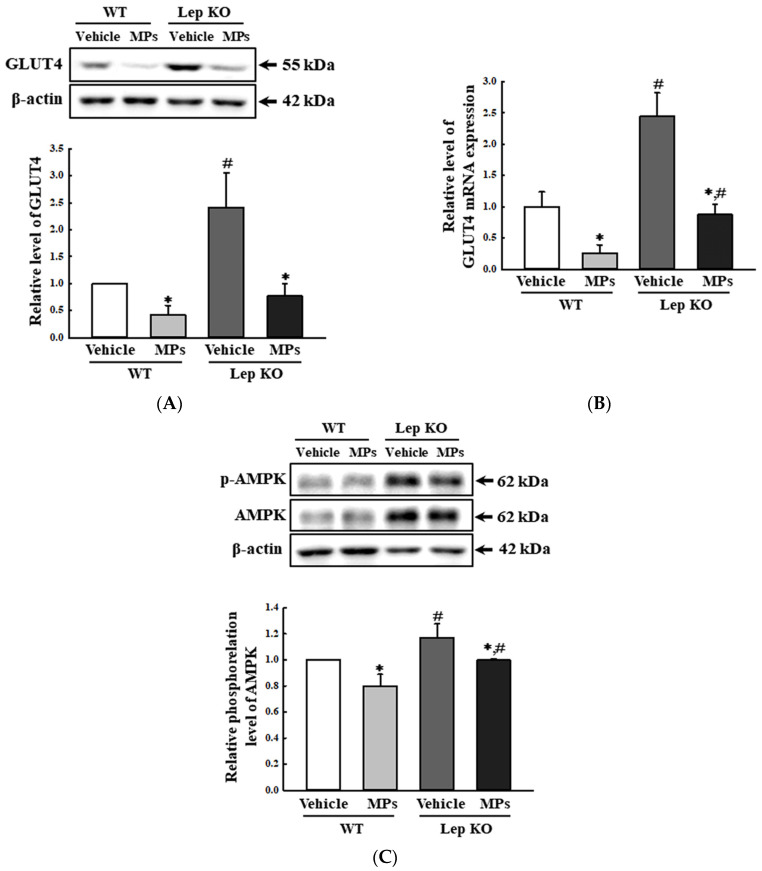
Level of key regulators in the GLUT4-AMPK signaling pathway. (**A**) Expression level of the GLUT4 protein. (**B**) Transcription level of the GLUT4 gene. (**C**) Phosphorylation level of AMPK. After the collection of the total proteins from the liver tissue, the expression levels of the GLUT4, p-AMPK, and AMPK proteins were determined by a Western blot. The preparation of the total homogenates from the liver tissue was performed for five to six mice per group, and the expression levels of each protein were analyzed twice for each sample. After the collection of the total RNA from the liver tissue, the mRNA level of the gene encoding GLUT4 was analyzed by RT-qPCR. The preparation of the total RNA from the liver tissue was performed on five to six mice per group, and the transcription levels of each gene were analyzed twice for each tissue. The data represent the means ± SD. * indicates *p* < 0.05 compared to the vehicle-treated group. ^#^ indicates *p* < 0.05 compared to the WT mice. Abbreviation: WT; wild type, Lep KO; leptin knockout, MPs; microplastics, RT-qPCR; reverse transcription-quantitative polymerase chain reaction, GLUT4; Glu transporter type 4, AMPK; 5′ AMP-activated protein kinase.

**Figure 12 ijms-25-04964-f012:**
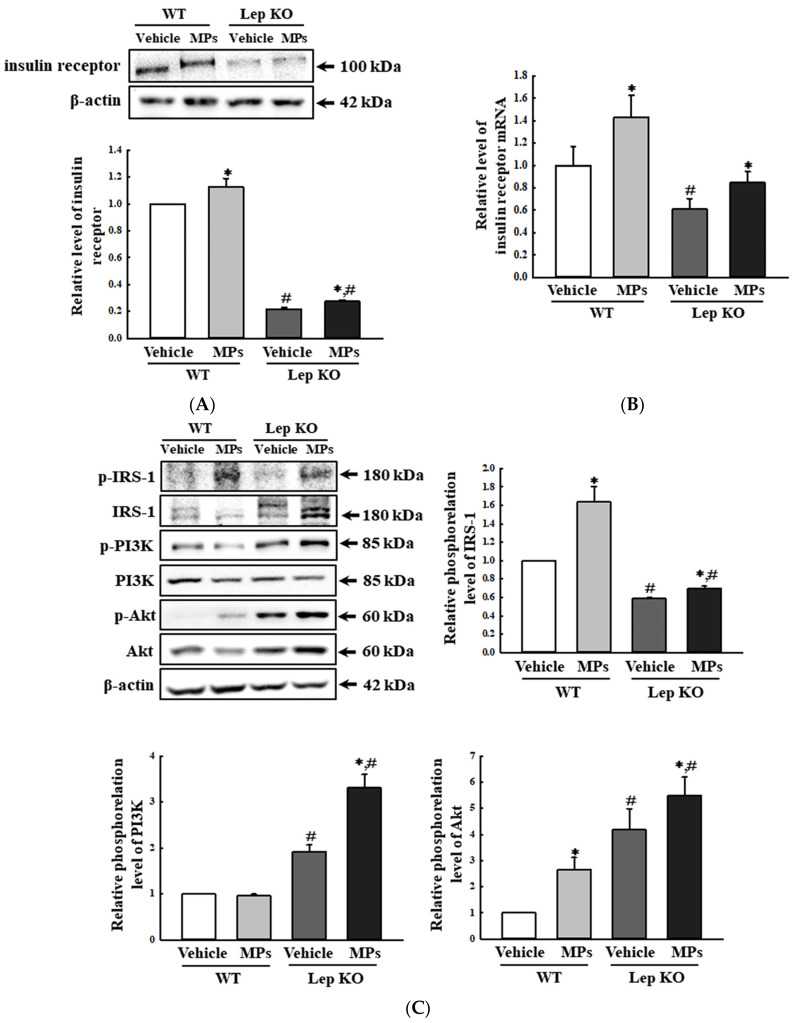
Expression levels of insulin receptors and key members of the insulin receptor downstream signaling pathway. (**A**) Expression level of the insulin receptor protein. (**B**) Transcription level of the gene encoding the insulin receptor. (**C**) Expression levels of the key mediators in the insulin receptor signaling pathway. After the collection of the total proteins from the liver tissue, the expression levels of insulin receptor proteins IRS-1, p-IRS-1, PI3K, p-PI3K, Akt, and p-Akt were determined by a Western blot. The preparation of the total homogenates from the liver tissue was performed on five to six mice per group, and the expression levels of each protein were analyzed twice for each sample. After the collection of the total RNA from the liver tissue, the mRNA level of the gene encoding the insulin receptor was analyzed by RT-qPCR. The preparation of total RNA from the liver tissue was performed on five to six mice per group, and the transcription levels of each gene were analyzed twice for each tissue. The data represent the means ± SD. * indicates *p* < 0.05 compared to the vehicle-treated group. ^#^ indicates *p* < 0.05 compared to the WT mice. Abbreviations: WT; wild type, Lep KO; leptin knockout, MPs; microplastics, RT-qPCR; reverse transcription-quantitative polymerase chain reaction, IRS-1; insulin receptor substrate 1, PI3K; phosphatidylinositol 3-kinase, Akt; protein kinase B.

**Figure 13 ijms-25-04964-f013:**
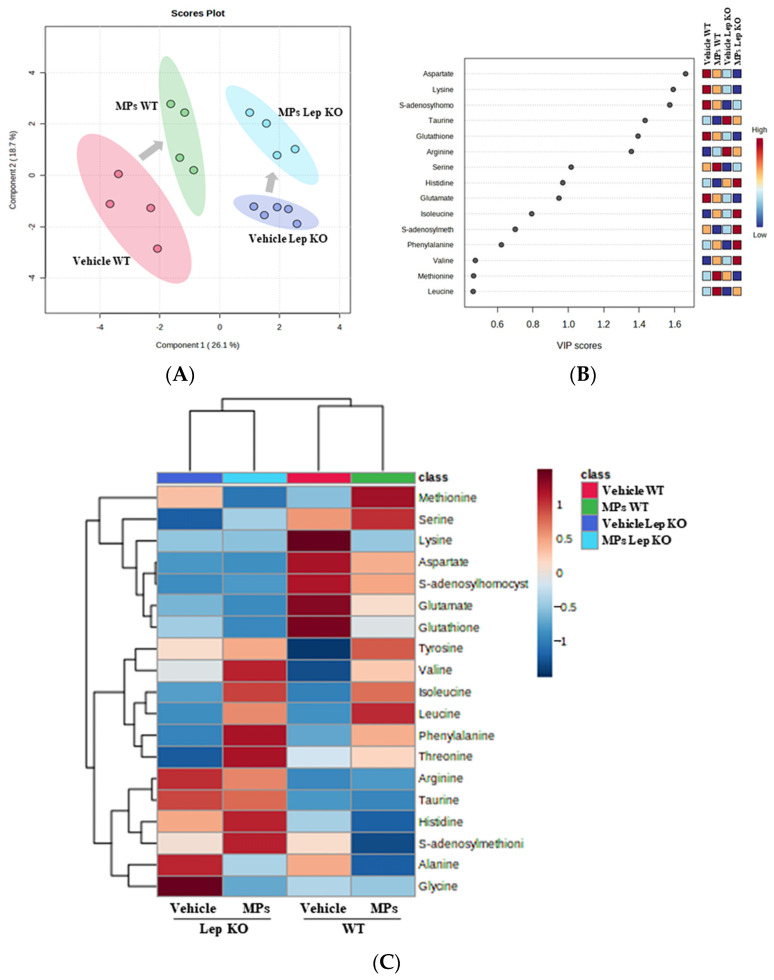
Spectral analysis of metabolite patterns using PLS-DA. (**A**) PLS-DA score plots comparing vehicle-treated groups and MP-treated groups. (**B**) VIP plot. The VIP plot shows the major metabolites contributing to cluster separation. (**C**) Heat map. Red color indicates a high concentration and blue color indicates a low concentration of endogenous metabolites. Abbreviations: WT; wild type, Lep KO; leptin knockout, MPs; microplastics, PLS-DA; partial least squares discriminant analysis, VIP; variable importance in projection.

**Figure 14 ijms-25-04964-f014:**
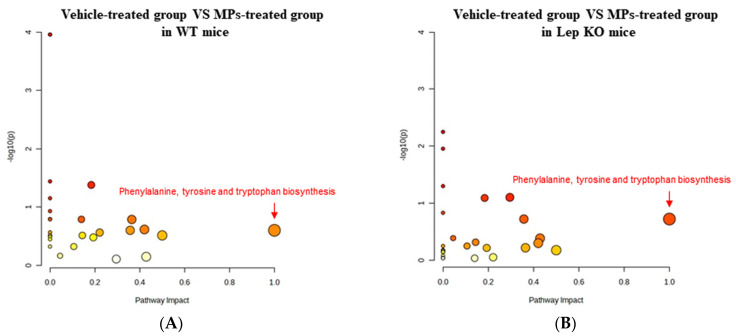
Metabolic pathway analysis of the endogenous metabolites in liver tissue. (**A**) Comparison between the vehicle- and MP-treated groups of WT mice. (**B**) Comparison between the vehicle- and MP-treated groups of Lep KO mice. The changes from yellow to red and from small to large circles indicate the amino acid metabolism pathways that showed a large response to MP administration. Abbreviations: WT; wild type, Lep KO; leptin knockout, MPs; microplastics.

**Figure 15 ijms-25-04964-f015:**
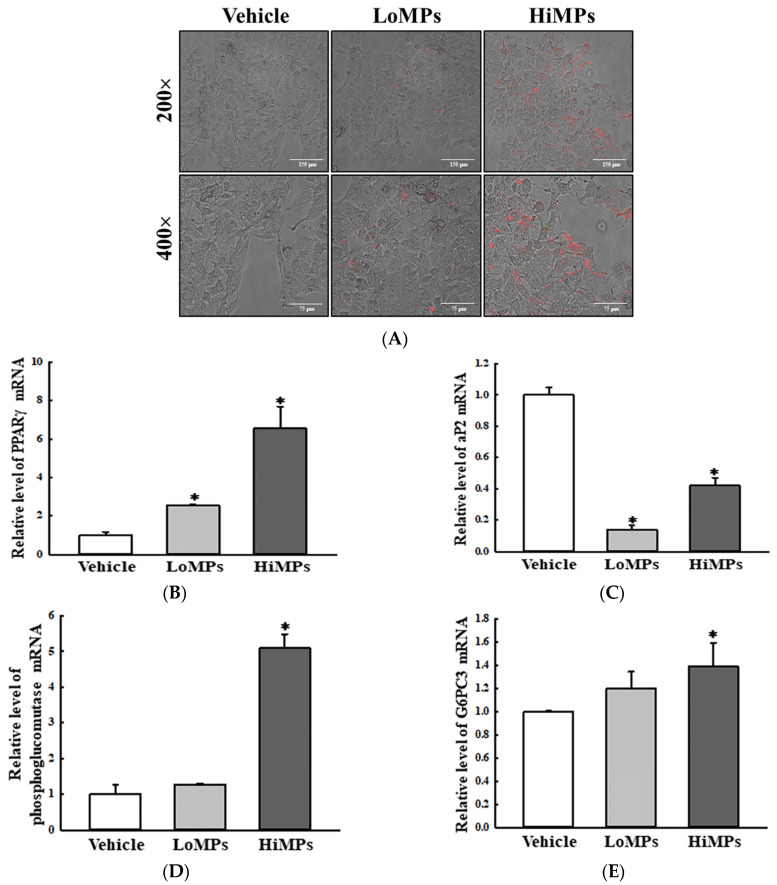
The transcription levels of the genes encoding adipogenic factors and glycogenolysis enzymes in HepG2 cells. (**A**) Morphology of the HepG2 cells after the MPs treatment. The red fluorescence of the MPs was detected under a fluorescence microscope at 200× and 400× magnification. (**B**,**C**) The transcription levels of the genes encoding adipogenic and lipogenic factors. (**D**,**E**) The transcription levels of the genes encoding the glycogenolysis enzymes. After the collection of total RNA from the HepG2 cells, the mRNA levels of the genes encoding PPARγ, aP2, phosphoglucomutase, and G6PC3 were analyzed by RT-qPCR. The preparation of total lysate and RNAs from the cells was performed on three to five wells per group, and the levels of the PCR product were analyzed twice for each sample. The data represent the means ± SD. * indicates *p* < 0.05 compared to the vehicle-treated group. Abbreviations: LoMPs; low-concentration microplastics, HiMPs; high-concentration microplastics, RT-qPCR; reverse transcription-quantitative polymerase chain reaction, PPARγ; peroxisome-proliferator activator receptor γ, aP2; adipocyte protein 2, G6PC3; Glu-6-phosphatase catalytic subunit 3.

**Figure 16 ijms-25-04964-f016:**
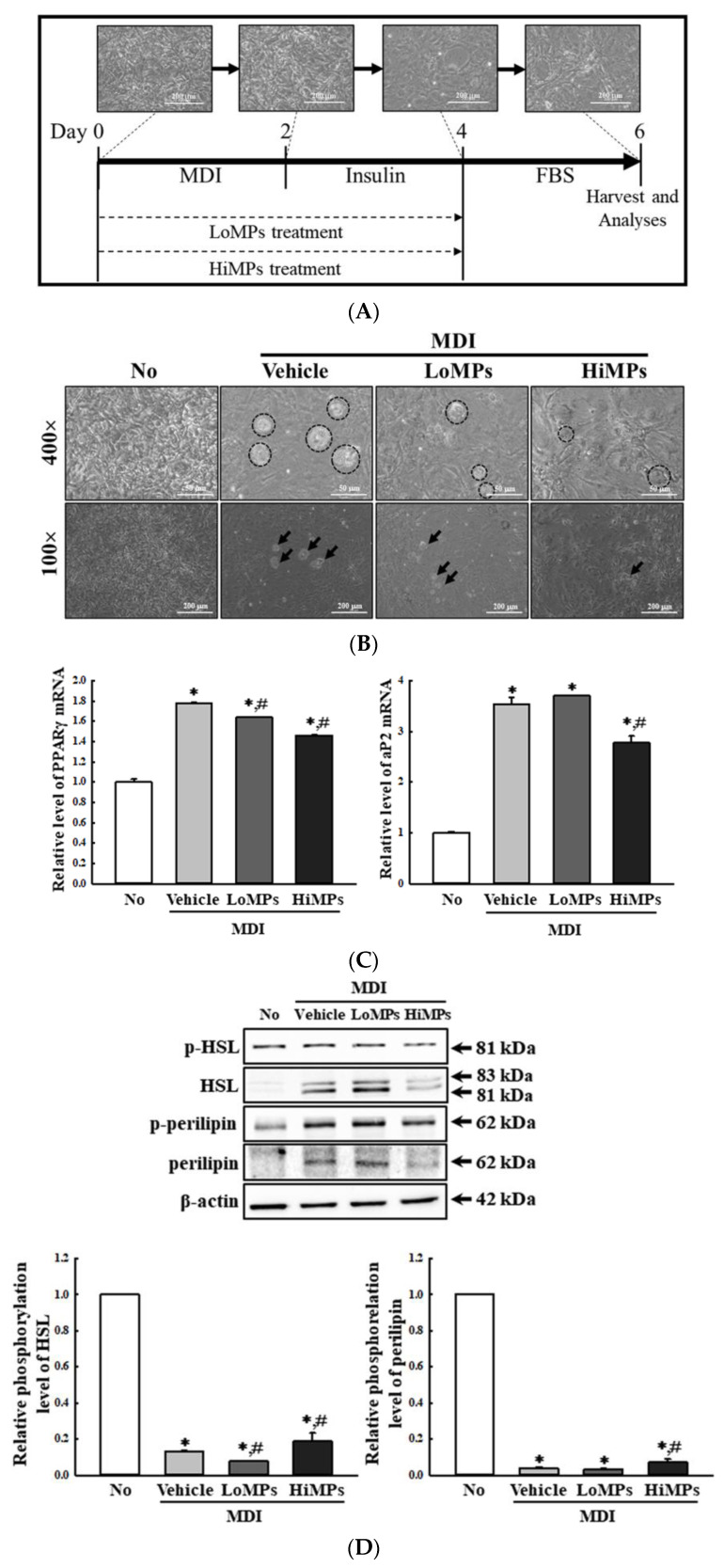
Levels of the adipogenic and lipogenic regulators in MDI-stimulated 3T3-L1 adipocytes. (**A**) Schedules for cell differentiation and MPs treatment. (**B**) Lipid accumulation of MDI-stimulated 3T3-L1 adipocytes after the MPs treatment. The image was detected at 100× and 400× magnification. Circles and arrows indicate mature adipocytes. (**C**) Transcription levels of the genes encoding adipogenic and lipogenic factors. (**D**) Phosphorylation levels of lipolysis factors. After the collection of total RNA from the 3T3-L1 cells, the mRNA levels of the gene encoding PPARγ and aP2 were determined by RT-qPCR. The preparation of total RNAs from the cells was performed on three to five wells per group, and the levels of each PCR product were analyzed twice for each sample. After the collection of the total proteins from the 3T3-L1 cells, the expression levels of four lipolysis factors (p-HSL, HSL, p-perilipin, and perilipin) were analyzed by a Western blot. The preparation of the lysates was performed on three to five wells per group, and the expression level of each protein was analyzed twice for each sample. The data represent the means ± SD. * indicates *p* < 0.05 compared to the No group. ^#^ indicates *p* < 0.05 compared to the vehicle-treated group. Abbreviations: LoMPs; low-concentration microplastics, HiMPs; high-concentration microplastics, RT-qPCR; reverse transcription-quantitative polymerase chain reaction, MDI; 3-isobutyl-1-methylxanthine, dexamethasone and insulin, FBS; fetal bovine serum, PPARγ; peroxisome-proliferator activator receptor γ, aP2; adipocyte protein 2, HSL; hormone-sensitive lipase.

**Figure 17 ijms-25-04964-f017:**
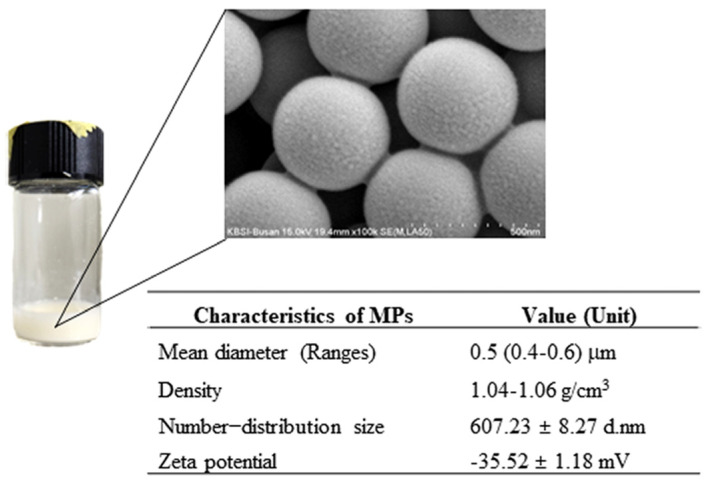
Morphology and physicochemical characteristics of MPs. These properties of MPs were analyzed as described in previous studies [50]. Data are reported as the mean ± SD. Abbreviation: MPs; microplastics.

**Figure 18 ijms-25-04964-f018:**
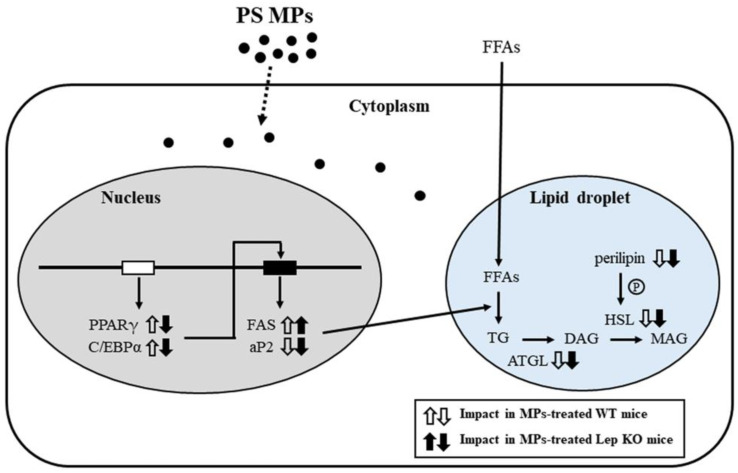
Schematic diagram of the mechanisms of MPs’ impacts on the liver. In this scheme, the administration of MPs is thought to disrupt lipid accumulation, adipogenesis, lipogenesis, and the lipolysis pathway in the liver. Abbreviations: PS MPs; polystyrene microplastics, PPARγ; peroxisome-proliferator activator receptor γ, C/EBPα; CCAAT/enhancer binding protein α, FAS; fatty acid synthase, aP2; adipocyte protein 2, FFAs; free fatty acids, TG; triglyceride; ATGL; adipose triglyceride lipase, DAG; diacylglycerol, HSL; hormone-sensitive lipase, MAG; monoacylglycerol, WT; wild type, Lep KO; leptin knockout.

**Table 1 ijms-25-04964-t001:** Lipid profile in the serum of WT and Lep KO mice after the administration of MPs.

Category	WT		Lep KO	
	Vehicle	MPs	Vehicle	MPs
TC (mg/dL)	73.25 ± 9.07	90 ± 5.66 *	133.67 ± 3.79 ^#^	142.5 ± 9.20 ^#^
TG (mg/dL)	50.75 ± 6.60	95 ± 19.80 *	39.75 ± 5.56	45 ± 4.24 ^#^
HDL (mg/dL)	63 ± 5.57	70.5 ± 12.02	99.67 ± 5.51 ^#^	107.5 ± 4.95 ^#^
LDL (mg/dL)	4 ± 1	8.5 ± 0.71 *	11 ± 3.2 ^#^	15.5 ± 3.54 ^#^

The data represent the means ± SD. * indicates *p* < 0.05 compared to the vehicle-treated group. ^#^ indicates *p* < 0.05 compared to the WT mice. Abbreviations: WT; wild type, Lep KO; leptin knockout, MPs; microplastics, TC; total cholesterol, TG; triglycerides, HDL; high-density lipoprotein, LDL; low-density lipoprotein.

## Data Availability

All the data that support the findings of this study are available on request from the corresponding author.

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
