# Peer review of "Impact of the Oral Administration of Polystyrene Microplastics on Hepatic Lipid, Glucose, and Amino Acid Metabolism in C57BL/6Korl and C57BL/6-Lepem1hwl/Korl Mice"

_ijms, 2024, doi:10.3390/ijms25094964_

Round 1

Reviewer 1 Report

Comments and Suggestions for Authors

The manuscript written by Roh et al describes the effects of microplastics in hepatic metabolism. The results are conclusive. However, minor improvements should be done prior publication:

Fig 15: The effects are hardly visible on the microphotographs. Please provide a higher magnification as supplementary figure.

Statistics: it should be mentioned if the data was tested regarding normal distribution.

One-sided or two-sided unpaired t-test?

Comments on the Quality of English Language

ok

Reviewer 2 Report

Comments and Suggestions for Authors

In the manuscript ID IJMS-2928278 entitled “Impact of the Oral Administration of Polystyrene Microplastics on Hepatic Lipid, Glucose, and Amino Acid Metabolism in C57BL/6Korl and C57BL/6-Lepem1hwl/Korl Mice”, the authors investigated the effects of MPs administration on hepatic metabolism in normal and obese mice and demonstrated its association with disruption of lipid, glucose, and amino acid metabolism. While some of the content is interesting, some of the content is very superficial and not discussed in depth.

There are some concerns about the manuscripts:

1.      The authors reported the impact of polystyrene microplastics on the liver using two types of mice. The authors stated that 9 weeks of oral administration of MPs is associated with the destruction of lipid, glucose, and amino acid metabolism in the liver tissues of obese WT and Lep KO mice. However, the pattern of hepatic steatosis appears to be reversed, making it difficult to understand the content, and the mechanism by which MPs induce these changes is complex and unclear. It is suggested that adding a schematic diagram of the mechanisms of influence on the liver, including mechanisms affecting PPARγ and lipid-related genes, would make it easier to understand.

2.      In Fig1, the localization of MPs in the liver is shown, but are MPs present within hepatocytes? Or are they present within blood vessels such as sinusoids? Or are they taken up by Kupffer cells? How long do MPs exist in the liver? Where do they ultimately go?

3.      In Fig15 and Fig16, the effects of MPs are evaluated using HepG2 and 3T3-L1 cells, but are MPs taken up by these cells? Or do MPs in the extracellular environment in the culture medium exert an influence? If the latter case, please explain how the influence of MPs occurs through which pathways.

Reviewer 3 Report

Comments and Suggestions for Authors

1.       This is interesting work, there are several questions and points requiring revision as below.

2.       There is the possibility that particle size of MP affects the phenotype of steatosis. The reason why the authors chose the size (0.4–0.6 μm) and the dose of MPs should be clearly described with comparing the experimental condition of previous studies.

3.       Why are the effects of MP administration so inconsistent between previous studies as described in introduction? No information about it makes the objective of this study uncomprehensive since elucidation of the mechanism is based on confirmed phenotype. There is no evidence that the phenotypes shown in this study are evolutionally confirmed ones compared to those of other groups. In essence, what had been not clarified in previous studies and what was done to uncover such unknown things including the strategy to overcome the defaults of previous studies should be logically and clearly described in introduction.

4.       Figure1, accumulation of MPs in the liver should quantify or calculate the percentage of the mouse which showed as a graph or table.

5.       Raw data of western blotting should be whole membrane in general. Parts of membrane is not good.

6.       Discussion is redundant. The description of discussion should not overlap with the content of introduction. In addition, the novelty and universality of the obtained results should be discussed with comparing previous studies.

7.       The potential molecular mechanism how MP affects fat accumulation should be discussed.

8.       Legend of figure 4, line 162 (aP2 and FAS) should be revised.

9.       Resolution of figure should be higher to make the letters clear. Different colors of lines also need to line graph to make it easy to distinguish each group.

Comments on the Quality of English Language

Minor editing of English language required

Round 2

Reviewer 2 Report

Comments and Suggestions for Authors

My concerns were cleared. Thank you

Reviewer 3 Report

Comments and Suggestions for Authors

OK

Comments on the Quality of English Language

As mentioned